# The Role and Mechanism of Metformin in the Treatment of Nervous System Diseases

**DOI:** 10.3390/biom14121579

**Published:** 2024-12-10

**Authors:** Hui Li, Ruhui Liu, Junyan Liu, Yi Qu

**Affiliations:** 1Key Laboratory of Birth Defects and Related Diseases of Women and Children (Ministry of Education), NHC Key Laboratory of Chronobiology, Department of Pediatrics, West China Second University Hospital, Sichuan University, Chengdu 610041, China; lihui980608@scu.edu.cn (H.L.); liujunyan1986@alu.scu.edu.cn (J.L.); 2Department of General Internal Medicine, West China Second University Hospital, Sichuan University, Chengdu 610041, China; liuruhui@gmail.com

**Keywords:** metformin, nervous system diseases, therapeutic effect, pharmacological mechanism

## Abstract

Nervous system diseases represent a significant global burden, affecting approximately 16% of the world’s population and leading to disability and mortality. These conditions, encompassing both central nervous system (CNS) and peripheral nervous system (PNS) disorders, have substantial social and economic impacts. Metformin, a guanidine derivative derived from a plant source, exhibits therapeutic properties in various health conditions such as cancer, aging, immune-related disorders, polycystic ovary syndrome, cardiovascular ailments, and more. Recent studies highlight metformin’s ability to cross the blood–brain barrier, stimulate neurogenesis, and provide beneficial effects in specific neurological disorders through diverse mechanisms. This review discusses the advancements in research on metformin’s role and mechanisms in treating neurological disorders within both the central and peripheral nervous systems, aiming to facilitate further investigation, utilization, and clinical application of metformin in neurology.

## 1. Introduction

The history of metformin can be traced back to 1922 when the compound was first synthesized by Irish scientists Emil Werner and James Bell. In 1957, it was first clinically applied to treat diabetes by French scientists. Metformin lowers blood glucose primarily by inhibiting hepatic glycogen synthesis and is the first-line treatment for type 2 diabetes. Its important functions in the body include downregulating blood glucose level, weight loss, anti-aging, anti-inflammation, the amelioration of polycystic ovary syndrome, and cardiovascular protection.

Metformin was pharmacologically optimized from a natural product isolated from the French clove (*Galega officinalis*) extract [1]. As one of the most common hypoglycemic drugs, metformin is recommended as a first-line medication for newly diagnosed type 2 diabetes in patients in many guidelines. It has long-term safety and efficacy, a low risk of hypoglycemia, cardiovascular benefits, mortality benefits, low cost, and wide availability [2,3]. Meanwhile, an increasing number of studies have shown the potential of re-purposing metformin to improve the treatment efficacy of cancer, immune-mediated diseases, aging, polycystic ovary syndrome, and other conditions [3,4,5,6,7]. Furthermore, it has also been reported to cross the blood–brain barrier and accumulate in the brain in vivo [8]. Recent studies have revealed that metformin has neuroprotective effects in the animal models of various nervous system diseases, including multiple sclerosis [9,10,11], stroke [12,13,14,15], Alzheimer’s disease [16,17,18,19,20], and so on. These neuroprotective effects are mainly attributed to rapid penetration across the blood–brain barrier and the pharmacodynamic activity of metformin [21,22,23,24]. By regulating oxidative stress, neuronal apoptosis, and inflammatory responses, metformin has potential therapeutic effects for a variety of neurological diseases [25,26,27]. Here, we systematically summarize the research progress on the role and mechanism of metformin in the treatment of central nervous system (CNS) and peripheral nervous system (PNS) diseases, including hypoxic–ischemic encephalopathy (HIE), stroke, Parkinson’s disease (PD), Alzheimer’s disease (AD), sepsis-associated encephalopathy (SAE), brain trauma, epilepsy, and different kinds of peripheral neuropathy.

## 2. Role and Mechanism of Metformin in CNS Diseases

The CNS has limited repair and neuronal regeneration ability [27,28,29,30], which restricts the efficacy of conventional therapies for CNS in the clinic. Cell replacement therapy has been proposed as a potential new treatment strategy for CNS diseases [30,31]. Metformin can facilitate the regeneration of neural precursor cells and promote their differentiation toward neurons and glial cells, thereby maintaining the quantity of neurocytes [32,33,34]. Currently, metformin is confirmed to positively contribute to the treatment of the following CNS diseases.

### 2.1. Metformin and Hypoxic–Ischemic Encephalopathy

Neonatal hypoxic–ischemic encephalopathy (HIE) is a cerebral hypoxic–ischemic injury caused by perinatal asphyxia resulting from maternal, fetal, placental, and other factors [35]. The asphyxia leads to cerebral ischemia (decreased cerebral blood flow) and hypoxia (decreased cerebral oxygen supply), which contribute to the basic pathological process of HIE. The primary pathological changes in HIE depend on the degree of brain maturation, as well as the severity and duration of the injury. Following hypoxic–ischemic events, the brain may exhibit edema, softening, bleeding, and necrosis, leading to the formation of voids, enlarged ventricles, and subdural hemorrhage. HIE-related brain injuries also lead to significant neurovascular inflammatory responses [36,37,38,39] and early structural changes in the vascular system [40]. Hypothermia therapy has been widely established as an effective treatment for HIE due to its effectiveness in reducing morbidity and disability [41,42]. However, it only offers a moderate degree of neuroprotective effects and is only effective for full-term newborns if treatment is started within six hours after HIE onset [43,44]. More effective treatment strategies are urgently needed to prevent or reduce neonatal brain injury after HIE.

In the mouse model of HIE, metformin increases the proliferation and the number of neural stem cells in the subventricular region of the lateral ventricle [45,46,47], and this effect is not influenced by gender [48]. Neural stem cells can differentiate into neurons and oligodendrocytes [49,50,51,52,53,54]. Accordingly, this effect of metformin can upregulate cell proliferation and differentiation in the CNS, counteracting brain damage in HIE (Figure 1). In other mouse and dog HIE models, metformin improves cognitive disorders by ameliorating myelin regeneration [55]. Therefore, metformin could be a promising drug for improving HIE treatment efficacy through multiple mechanisms.

In addition, different drug concentrations and dosing schedules influence the effects of metformin on cerebral ischemic damage. In the acute phase of cerebral infarction, metformin exacerbates acute brain injury in a rat model of cerebral ischemia [56]. However, the long-term administration of metformin reduces the volume of brain atrophic parts and promotes the recovery of ethological indicators [57,58,59] (Figure 1). Similar effects also occur in the mouse model. When hypoxic–ischemic modeling starts, ischemia-induced apoptosis is effectively inhibited by metformin administration at 10 mg/kg/day for seven days in advance. Interestingly, the inhibitory effect is weakened if administered only three days in advance. Furthermore, metformin administration in the acute phase of an acute ischemic injury model increases the cerebral infarction volume [60]. Therefore, metformin administration should be avoided during the acute phase and instead be administered during the stable phase or prophylactically.

### 2.2. Metformin and Stroke

Stroke is divided into ischemic and hemorrhagic types, with thromboembolism being the main cause [61,62,63]. It leads to an inflammatory microenvironment, neuronal death, and neurological defects [64,65]. In rat stroke models, metformin promotes microglia to divert from the M1 to M2 phenotype, alleviates the inflammatory microenvironment, and directly inhibits glutamate-induced neuronal excitotoxicity, thereby facilitating brain tissue repairment [66,67] (Figure 1). At the molecular level, metformin downregulates apoptotic factors (p-JNK3, p-c-Jun, and caspase-3) and pro-inflammatory cytokines (IL-1β, IL-4, IL-6, and TNF-α) to mitigate stroke damage in rat models [68] (Figure 1).

### 2.3. Metformin and Parkinson’s Disease

Parkinson’s disease (PD) is a progressive neurodegenerative disease caused by the loss of dopamine neurons in the substantia nigra and striatum, manifested by muscle rigidity, bradykinesia, static tremors, and postural instability [69,70,71,72,73]. Currently, metformin is simultaneously reported to ameliorate and aggravate PD in different studies, which is controversial. Metformin inhibited neuronal apoptosis and neuroinflammation by activating the AMP-activated protein kinase (AMPK)-autophagy signaling pathway, alleviated dopamine neuron injury, and provided a neuroprotective effect for PD models in some studies [74] (Figure 1). However, other studies showed that metformin reduced the activation of microglia and aggravated dopaminergic neuron damage rather than being protective [75]. Well-designed experiments are required to further confirm whether metformin could have either protective or harmful effects on PD neuropathology, as there are discrepancies between studies in demographics, drug dosage and duration, drug categories, follow-up periods, and adjusted variables [76].

### 2.4. Metformin and Alzheimer’s Disease

Alzheimer’s disease (AD) is a neurodegenerative disease that causes memory loss and cognitive disorders. Its main histopathological feature is amyloid-β (Aβ) plaque deposition in brain tissue, surrounded by microglia and astrocytes, with neurofibrillary tangle formation and the loss of neurons [77,78,79,80,81]. At present, controversial reports exist regarding the cognition improvement effect of metformin. Some studies show that metformin is beneficial for AD. Metformin activates AMPK to inhibit Aβ deposition, improves mitochondrial function, and inhibits oxidative stress in an AD mouse model (Figure 1). It decreases hippocampal neuron death and promotes their regeneration, improves AD-related neuropathological changes, and alleviates cognitive impairment [82] (Figure 1). In terms of clinical studies, a trial in a Taiwan province of China using a randomized controlled trial investigated patients who took non-metformin hypoglycemic drugs orally. After an 8-year follow-up, the risk of dementia in type 2 diabetes mellitus patients was found to be more than doubled compared with the control group, and the oral administration of metformin reduced this risk [83]. A similar trial in 2014 reached a similar conclusion that metformin could lower the risk of dementia in type 2 diabetes mellitus patients [84]. In another parallel randomized controlled clinical trial, patients with mild cognitive impairment but without a clear diagnosis of diabetes were randomized into a metformin-treated group and a placebo group. Metformin treatment was found to improve performance in the selective recall test by implementing cognitive function changes after medication [85].

On the other hand, metformin has also been reported to exacerbate AD. AMPK signaling activated by metformin is reported to promote the accumulation of Aβ and aggravate cognitive disorders in male AD rat models, suggesting that metformin promotes the development of AD in rats [86]. A retrospective pathological control study showed that the long-term oral administration of metformin led to vitamin B12 deficiency, which is related to deterioration in cognitive performance [87,88]. Another study showed that vitamin B12 and calcium supplements might alleviate metformin-induced vitamin B12 deficiency and improve cognitive functions [89], indicating that metformin is associated with deterioration in cognitive functions.

Analyzing these contradictory conclusions, we found differences in the study populations between these clinical trials, specifically in terms of race. Metformin-mediated memory improvement mainly occurred in Asian populations, whereas the aggravation mainly occurred in white populations. Additionally, and interestingly, gender may also be an influencing factor. Metformin is found to aggravate memory impairment in male mice but offers benefits to female mice. These findings suggest that race and gender could be important factors influencing the treatment effects of metformin in AD. Generally, metformin is mainly reported to exert positive effects on cognitive function in the AD context, combining the results of experimental models and clinical studies. In many cases, it reduces the inflammatory indicators of AD and decreases the quantity of Aβ and phosphorylated Tau proteins (Figure 1).

### 2.5. Metformin and Sepsis-Associated Encephalopathy

Sepsis-associated encephalopathy (SAE) is a systemic inflammatory response-mediated brain dysfunction, which increases the incidence of cognitive impairment, seriously affecting patient prognosis [90,91]. The pathogenesis of SAE mainly includes the disturbance of cerebral microcirculation, damage to the blood–brain barrier, the activation of oxidative stress, inflammatory response, and neuronal apoptosis [92]. SAE is associated with learning and memory disorders due to its CNS damage that mainly occurs in the hippocampus [90]. Metformin attenuated oxidative stress in the hippocampus, thus alleviating learning and memory disorders in mice [93,94] and rat [95] SAE models (Figure 1). Metformin also activated PI3K/Akt signaling and induced anti-inflammatory effects to elicit brain protective effects in a mouse SAE model [96] (Figure 1).

### 2.6. Metformin and Traumatic Brain Injury

Traumatic brain injury (TBI) is caused by traumatic mechanical force and involves complex pathophysiological processes, including oxidative stress, inflammation, mitochondrial dysfunction, and apoptosis [97]. It includes cerebral contusion, diffuse axonal injury, and intracranial hemorrhage resulting from mechanical impact and has two main stages of primary and secondary injuries [97,98,99,100,101]. Metformin can activate NF-κB and MAPK signaling to inhibit inflammatory responses, offering therapeutic benefits in rat TBI models [102,103] (Figure 1). It also activates the Ser/Thr kinase family member Par1, regulating synaptic plasticity and neuroinflammation to improve cognitive function and spatial learning [68] (Figure 1). In addition, clinical trials have shown that metformin may be an effective and safe treatment intervention for severe TBI patients [104].

### 2.7. Metformin and Epilepsy

Epilepsy is a brain dysfunction syndrome associated with cognitive, sensory, and (or) motor dysfunction caused by the abnormal discharge of neurons in the brain [105]. Recent findings indicate that oxidative stress and mitochondrial dysfunction are found to be closely related to the occurrence and development of epilepsy and cognitive impairment [106]. Metformin can impede the inflammatory process in epileptic rat models to improve cognitive function, offering potential neuroprotective advantages [107]. At the molecular level, metformin reduces brain oxidative damage, activates AMPK, inhibits the mTOR pathway, downregulates α- synuclein, and alters the levels of brain-derived neurotrophic factors and TrkB to downregulate apoptosis, thereby mediating antiepileptic effects [108,109] (Figure 1). The mechanism of metformin in the treatment of central nervous system diseases is summarized in Figure 1.

## 3. Role and Mechanism of Metformin in PNS Diseases

The PNS is mainly made up of neurons and their cellular extensions. Its main function is to connect peripheral receptors with the CNS [110]. Metformin has been found to provide advantages in PNS diseases, especially different kinds of peripheral neuropathy.

### 3.1. Metformin and Peripheral Neuropathic Pain

The peripheral nerves are functionally divided into two components: sensory input and motor output. The former comprises the spinal nerve’s posterior root, the posterior root ganglion, and the sensory nerves of the brain. Peripheral nerve fibers are classified into two types: myelinated and unmyelinated. Nerve fibers form the fundamental building blocks of peripheral nerve structures, with many fibers grouped into nerve bundles and multiple bundles comprising the nerve trunk. The prevalence of pain in the peripheral nervous system is high, significantly affecting patients’ quality of life. Alleviating this crippling pain condition presents a challenge for healthcare professionals [111]. Experiments evaluated the risk of herpes zoster and postherpetic neuralgia between metformin users and non-users and found that the use of metformin significantly reduced the risk of herpes zoster and postherpetic neuralgia, and the higher the cumulative dose of metformin, the lower the risk of these disorders [111]. The analgesic effect of metformin on chronic constrictive injury (CCI)-induced neuropathic pain (NP) was investigated in a CCI rat model, and it was found that metformin activated AMPK and inhibited the activation of the STAT3 signaling pathway in CCI rats, thereby inhibiting the activation of astrocytes and microglia, and further inhibiting the release of inflammatory cytokines to alleviate NP [112].

### 3.2. Metformin and Diabetes Peripheral Neuropathy

Diabetes peripheral neuropathy is one of the most common chronic complications of diabetes. Specifically, when other reasons are excluded, diabetes patients have symptoms related to peripheral nerve dysfunction. The clinical manifestations are symmetric pain and sensory abnormalities. Lower limb symptoms are more common than upper limb symptoms. In a study exploring the effect of metformin on sciatic neuritis in insulin-dependent diabetes mice, the experimental results show that high-dose metformin prevented the atrophy of medullary axons and reduced the expression of inflammatory mediators (IL-1β, inducible nitric oxide synthase, and nitric oxide), thus protecting the peripheral nerve damage caused by chronic hyperglycemia [113]. SD rats were used to build diabetes models, and it was found that metformin improved the sciatic nerve morphology in rats with diabetes in terms of axon diameter, myelin sheath thickness, and myelinated fiber diameter [114]. In a diabetes rat model, authors found that metformin reduced the levels of malondialdehyde and advanced glycation end products in the blood and increased superoxide activity, thus reducing the hyperalgesia and abnormal pain induced by diabetes [115]. It found that metformin significantly weakened the anti-injury tolerance of morphine in diabetic rats; at the same time, it reduced neuronal apoptosis through downregulating pro-apoptosis caspase-3 and Bax while upregulating the anti-apoptosis protein Bcl-2 [116]. In another study, cold and hot stimulation tests were conducted on diabetic and non-diabetic mice. After detecting the levels of TNF–α, IL-6, and nitrite in the brain and liver tissues of the two groups of mice, it was found that the mice treated with the sildenafil–metformin combination increased the latency of the pain response in diabetic animals in a dose-dependent manner and reduced the concentration of nitrite in the brain and liver [117]. It was found that metformin alleviated the mechanical abnormal pain induced by diabetes, and the analgesic effect was blocked by compound C (an AMPK inhibitor). At the same time, it also enhanced the phosphorylation level of AMPK in the L4-6 dorsal root ganglia of diabetic rats and reduced the expression of NF-κB but did not affect the expression of total AMPK [118].

### 3.3. Metformin and Chemotherapy-Induced Peripheral Neuropathy

Chemotherapy-induced peripheral neuropathy (CIPN) is one of the most common causes for cancer patients to interrupt treatment in the early stages. Some patients can alleviate symptoms by reducing the dosage of chemotherapy or temporarily stopping chemotherapy. However, for other patients, symptoms can persist for months, years, or even a lifetime. There is currently no effective prevention method for peripheral neuropathy, and early detection and treatment are particularly important. CIPN progresses to glove and sock neuropathy, which, in more severe cases, can proximally spread to affect most limbs. Though it is primarily a sensory neuropathy, it also impacts the autonomic nervous function, as well as fine motor function and proprioception. Evidences suggest the loss of sensory fibers and reduced density of nerve fibers within the epidermis [119,120,121,122]. Metformin effectively prevented oxaliplatin-induced intraepidermal fibrosis in a CIPN rat model using SD rats with oxaliplatin intraperitoneal injection; it also inhibited the activation of astrocytes in the spinal cord and prevented mechanical and cold hypersensitivity reactions induced by chemotherapy [123]. The same is true with C57Bl/6J mice, where metformin prevented chemotherapy-induced neuropathic pain and numbness, increased cisplatin-induced latency, prevented sensory impairment, and inhibited the decrease in epidermal nerve fiber (IENF) density in the paws [124]. Oxaliplatin caused a nociceptive response in Swiss male mice, accompanied by the increased expression of c-Fos and ATF3 in the dorsal root ganglia and spinal cord. Metformin significantly attenuated oxaliplatin-related nociception, reduced the expression of c-Fos and ATF3 and prevented oxaliplatin-induced peripheral sensory neuropathy [118]. Metformin alleviated bortezomib-induced neuropathic pain by reducing the expression of the autophagy marker Beclin-1 in the spinal dorsal horn [111]. Moreover, some researchers evaluated the role of metformin in peripheral neuropathy in stage III colorectal cancer patients using a randomized controlled study and found that the average pain score of the metformin group was significantly lower than that of the control group, and the average serum levels of malondialdehyde and neurotensin were significantly reduced, indicating that metformin can protect colorectal cancer patients from oxaliplatin-induced chronic peripheral neuropathy [125].

### 3.4. Metformin and Nutritional Peripheral Neuropathy

Nutritional peripheral neuropathy occurs in two forms: an isolated deficiency (usually of a B vitamin) or a complex deficiency resulting from several concurrent metabolic disorders (usually including malabsorption). Causes of nutritional peripheral neuropathy include alcohol exposure, thiamine deficiency, niacin deficiency, pyridoxine deficiency and excess, cyanocobalamin deficiency, pantothenic acid deficiency, alpha-tocopherol deficiency, celiac disease, and multifactorial mechanisms. Manifesting symptoms like numbness, pain, and a burning sensation in the limbs can lead to damage in both the muscles and nerves, resulting in a decline in muscle strength. In extreme cases, muscle atrophy and a reduction in sensation may also be observed. C57BL/6 mice were used to construct a high-fat diet model, and it was found that treatment with metformin combined with chlorogenic acid promoted the expression of anti-inflammatory cytokines (IL-10) and inhibited the expression of pro-inflammatory mediators (TNF-α, IL-1β, MCP-1, and IL-6), reducing obesity-induced skeletal muscle inflammation response [126]. In the rare autosomal inherited disease of progressive fibrodysplasia, metformin can inhibit the osteogenic differentiation of Smad6 and Smurf1 in vitro, thereby reducing the development of ectopic bone formation in the muscles [127]. Metformin not only controls blood sugar but also reduces the expression of inflammatory markers (IκBα, SOD2, MCP1, p-ERK, and JNK) in muscles [128]. In another study, an experimental autoimmune myasthenia gravis rat model was used, and it was discovered that metformin potently suppresses Th17 cell differentiation through the elevation of reactive oxygen species and AMPK, rectifying the imbalance among various T-cell subsets, thereby modulating the progression of myasthenia gravis [129]. From a clinical perspective, data from the South Korean National Health Insurance were used in a study to conduct a retrospective cohort study involving patients with dyslipidemia who either were or were not concurrently treated with metformin. The results indicate that metformin can mitigate the risk of myopathy and afford protection against the potential muscle toxicity induced by statin therapy [130]. The role and mechanism of metformin in the treatment of nervous system diseases are summarized in Table 1.

**Table 1 biomolecules-14-01579-t001:** Role and mechanism of metformin in the treatment of nervous system diseases.

Disease Name	Animal Models or Clinical Trials	Concentration or Dose/In Vitro or In Vivo	Role and Mechanism	References
Hypoxic–ischemic encephalopathy	Mouse	300 mg/kg/day in drinking water; 50 mg/kg/day intraperitoneal; 200 mg/kg/day;	Promotes neural stem cells to differentiate into neurons and oligodendrocytes Improves cognitive ability by reducing myelin regeneration impairment; decreases the ischemia-induced apoptosis of cerebral cortex	[57,58,60,61,64,66,67,74]
Stroke	Mouse	300 mg/kg/day	Facilitates the transformation of microglia from the M1 to M2 phenotype; inhibits glutamate-induced neuronal excitotoxicity and death; decreases pro-apoptotic factors (p-JNK3, p-c-Jun and caspase-3) and reduces pro-inflammatory cytokines (IL-1β, IL-4, IL-6 and TNF-α)	[66,67,68]
Parkinson’s disease	Mouse	200 mg/kg/day; 20 μg/kg intraperitoneal	Activates AMPK-autophagy signaling pathway; alleviates neuronal injury	[86]
Alzheimer’s disease	Clinical trial		Decreases the risk of developing dementia in T2DM patients; improves the score of the selective reminiscent test in patients with mild cognitive disorders without diabetes	[83,84,85]
	Mouse	200 mg/kg/day	Activates AMPK, which can inhibit Aβ deposition; it improves mitochondrial function and inhibits oxidative stress; decreases hippocampal neuronal death; promotes hippocampal neuronal occurrence; improves Alzheimer’s disease-related neuropathological changes; reduces cognitive impairment	[89]
Sepsis-associated encephalopathy	Rat	100 mg/kg orally	Anti-inflammatory and anti-oxidant action; activates the PI3K/Akt signaling pathway	[104,107,108]
Peripheral neuropathic pain	Rat and clinical trial	200 mg/kg BW/day orally	Reduces the risk of shingles and postherpetic nerve pain; activates AMPK and inhibits the activation of the STAT3 signaling pathway, thereby inhibiting the activation of astrocytes and microglia	[111,112]
Diabetes peripheral neuropathy	Rat and mouse	100 mg/kg/day orally 200 mg/kg/day orally	Prevents the atrophy of myeloid axons and reduces the expression of inflammatory mediators; restores the shape of the sciatic nerve; alleviates mechanical hyperalgesia, thermal hyperalgesia, and abnormal cold pain caused by diabetes; downregulates the pro-apoptotic protein caspase 3 and Bax of the dorsal root ganglion; upregulates the level of anti-apoptotic protein Bcl-2	[114,115,116,117,118]
Chemotherapy-induced peripheral neuropathy	Rat and mouse	100 mg/kg/day orally 250 mg/kg/day orally 250 mg/intraperitoneal	Prevents intraepidermal fibrosis; inhibits spinal astrocyte activation and mechanical and cold hypersensitivity reactions; inhibits a decrease in the density of epidermal nerve in thepaws; reduces the expression of c-Fos and ATF3	[111,118,123,124,125]
Nutritional peripheral neuropathy	Rat and mouse	10 mM in vitro 10 mL/kg intraperitoneal	Reduces inflammation; inhibits osteogenic differentiation; reduces the expression of inflammatory markers in muscles; reduces the risk of myopathy and provides protection against potential muscular toxicity; inhibits Th17 cell differentiation and corrects imbalances among various T cell subpopulations	[126,127,128,129,130]

p-c-Jun N-terminal kinase 3, p-JNK3; interleukin, IL; tumor necrosis factor alpha, TNF-α; AMP-activated protein kinase, AMPK; Type 2 diabetes mellitus, T2DM; amyloid beta, Aβ; phosphoinositide 3-kinase, PI3K; Signal transducer and activator of transcription 3, STAT3; activating transcription factor 3, ATF3.

## 4. Conclusions and Perspectives

In this review, we summarized the role and mechanisms of metformin in neurological disorders (Figure 1 and Table 1). Metformin initiates autophagy, shields against oxidation, diminishes neuroinflammation, and functions as a neuroprotective and neurorestorative element across various neurological disorders. It also hinders neuroinflammation and apoptosis through the modulation of distinct signaling pathways like NF-κB and AMPK, consequently enhancing the pathological progression of neurodegenerative conditions. Furthermore, metformin stimulates neuronal regrowth and synaptic interconnections, thereby aiding in the recovery of neurological capabilities. Its potential in the treatment of neurological disorders has been demonstrated both in vivo and in vitro, providing a promising strategy for the treatment of related diseases in the future.

As the first choice for full-course treatment in type 2 diabetes, metformin has been widely used in clinical practice for decades. After oral administration, metformin is absorbed through the intestine into the bloodstream. It is rapidly cleared by the kidneys in its original form and has a weak but specific inhibitory effect on the respiratory chain complex of mitochondria in skeletal muscle and the liver, with no metabolism needed by the liver and no binding to proteins [131]. Metabolizing metformin in the body is generally considered safe for the majority of patients. Nevertheless, there is a possibility of heightened risk in particular circumstances, such as hepatic or renal impairment or in instances of overdose.

The widespread use of metformin has given rise to persistently accumulating epidemiological data, allowing us to explore its impact on conditions beyond diabetes. A growing body of research and evidence is ushering this clinically approved drug into the possibility of repurposing it for weight loss, cardiovascular disease, cancer, aging, polycystic ovary syndrome, as well as neurological disorders. Currently, new perspectives from clinical studies are being further validated in clinical trials and have achieved significant milestones. It is also important to identify the targeted pathways and molecules influenced by metformin. These molecules can be used as biomarkers to stratify subgroups of patients who exhibit different molecular or pathological characteristics and respond to metformin treatment, ultimately leading to the development of targeted therapies [10]. Subsequent research is still urgently needed to promote the application of metformin in neurological diseases and to offer even more benefits to relevant patients.

## Figures and Tables

**Figure 1 biomolecules-14-01579-f001:**
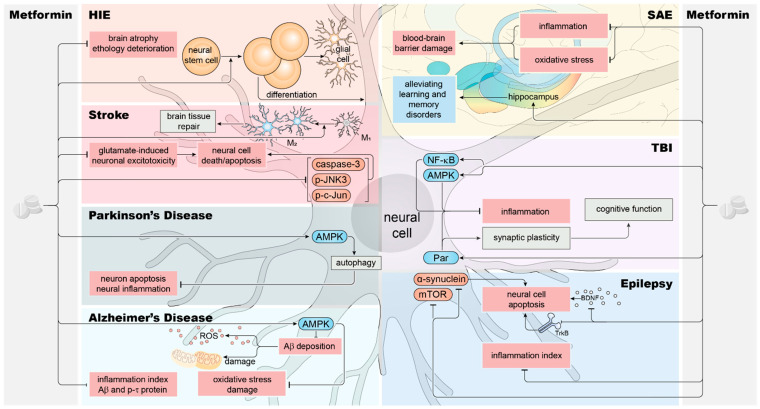
The mechanism of metformin in the treatment of central nervous system diseases. Solid arrows represent facilitation; T-shaped arrows represent inhibition.

## Data Availability

Not applicable.

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
