# Peer review of "The Role and Mechanism of Metformin in the Treatment of Nervous System Diseases"

_biomolecules, 2024, doi:10.3390/biom14121579_

Round 1
Reviewer 1 Report
Comments and Suggestions for Authors
I read with great interest the submitted paper, as the subject of potential repurposing of metformin for various conditions, including neurological disorders, is an area of extensive research of the latest decade.
The paper is a narrative review of studies investigating the potential benefits of metformin in hypoxic-ischemic encephalopathy, stroke, Parkinson's disease, Alzheimer's disease, sepsis-associated encephalopathy, brain trauma, epilepsy, and various peripheral neuropathies.
However, my opinion is that in its current form, the paper lacks the power of illuminating with synthetic information. I I support my statement by the following concerns and recommendations for improvement.
1. Abstract – include mostly general statements, I would recommend a more structured version, including the main findings of the study
2. The objective of the study is not clearly defined. Is it a systematic review? Of which type of studies? Experimental (in vitro, animal studies), clinical studies? It is not clear. Some references to clinical studies are made for some conditions, but not in all of them. Important references on the subject are not included
3. A methodology section should be included (search strategy, search terms, databases, etc).
4. The statement on raws 49-51 is not accurate: “Currently, metformin is confirmed to positively contribute to the treatment of the following CNS diseases.” – if it was confirmed, which would be the objective of this review?
5. In every subchapter there are different types of study mentioned, mixing results of in vitro/animal studies with results of clinical studies (no mention on study design, RCT or epidemiological studies). The level of evidence for each case should be presented separately.
6. An explanation of figure 1 would be necessary on how are all the elements connected
7. A section explaining how different effects (on molecular level) would be of benefit in the pathological condition analysed would be useful to the reader
8. Chapter 2.3 –scarce in information, No information on clinical studies. see article Metformin: The Winding Path from Understanding Its Molecular Mechanisms to Proving Therapeutic Benefits in Neurodegenerative Disorders (mdpi.com) for a detailed review of the studies and of the mechanisms involved in possible beneficial effects.
9. Chapter 3.4 – apart from general statements about neuropathies, the chapter enumerates various studies without pointing out the connection to the subject of nutritional neuropathies (i.e Metformin-chlorogenic acid combination reduces skeletal muscle inflammation in c57BL/6 mice on high- 583 fat diets, AMPK downregulates ALK2 via increasing the interaction between Smurf1 and Smad6, leading to inhibition of 585 osteogenic differentiation, Impact of metformin on statin-associated myopathy risks in dyslipidemia patients, and a study on possible beneficial effects of metformin in myasthenia gravis).
10. Conclusion section – is not based on the paper content. It includes: general information of metformin, such as pharmacokinetics. Inappropriately included in the paragraph of pharmacokinetics there is a mention on a pharmacodynamic effect, that on the respiratory chain complex: “It is rapidly cleared by the kidneys in its original form and has a weak but specific inhibitory effect on the respiratory chain complex of mitochondria in skeletal muscle and liver, with no metabolism by liver and binding to proteins”. There are mixed information on elimination, than mechanism of action, than metabolism and then protein binding – in a sequence which has no meaning. The paragraph ends with a statement of an adverse effect
- The next paragraph in Conclusion speaks about nanoparticles formulation of metformin, again, totally unrelated to the article title and content
I would recommend totally changing the conclusion section with the main results of the current study
Minor changes:
1. CHANGE metformin hypoglycaemic drug with antihyperglycemic drug
2. Chapter 3.4 – definition of Nutritional peripheral neuropathy is incorrect.
Nutritional nueropathies occur in two forms: an isolated deficiency (usually of a B vitamin) or a complex deficiency resulting from several concurrent metabolic disorders (usually including malabsorption). Causes of nutritional neuropathies include alcohol exposure, thiamine deficiency, niacine deficiency, pyridoxine deficiency and excess, cyanocobalamin deficiency, pantothenic acid deficiency, alpha-tocopherol deficiency, celiac disease, and multifactorial mechanisms. (Nutritional Neuropathy: Practice Essentials, Background, Pathophysiology (medscape.com))
Reviewer 2 Report
Comments and Suggestions for Authors
The study is very interesting but does not comply with the PRISMA criteria. The authors did not mention the methods, the criteria, used for the inclusion of the studies included in the study.
PRISMA criteria must be included if it is desired to increase the quality and importance of this study.
This review does not comply with the Prima criteria mentioned here: ttps://www.mdpi.com/journal/biomolecules/instructions. This in not a a systematic review.
It must be specified if a software was used to create figure 1 or where the information is included from.
Reviewer 3 Report
Comments and Suggestions for Authors
Reviewer:
In this review, the author summarizes the progress on metformin for central nervous system (CNS) and peripheral nervous system (PNS) diseases, and its potential mechanism. While the writing is very well, there are several critical points which need the author’s attention.
Specific points:
1. The author needs to briefly introduce the history of metformin including how it is found, and its important function in vivo.
2. Another part needs to ed added to systematically introducing the function of Metformin in neurons, glia cells etc., and its physiological role in the brain.
3. The disadvantages of metformin also should be introduced in the author’s review.
4. Line 21: Delete any extra spaces.
5. Line 23: Delete “1.”, Just introduction.
6. Figure 1 looks so complicated, more lines and arrow, and the author should make it simple.
7. The grammar and sentence need more check.
Reviewer 4 Report
Comments and Suggestions for Authors
The submitted manuscript aimed to summarize the current knowledge on metformin in therpay of CNS disorders. Albeit interesting, the mnuscript lacks consistency. Therefore, the following changes must be introduced prior to the submission:
1) in abstract, an info that review is of a narrative type should be added
2) the content - the authors did not use consistent way of presenting the data coming from either experimental or clinical experiments
a) experimental - please, add the concentrations used in studies in vitro as well as doses used in vivo
b) for clinical data - please, add the number of enrolled patients for all mentioned studies or remove these data from paragraph 2.4.
c) the names of scientists are usually mentioned in the text only when their contribution to science is unique and substantial (first and improatnt discovery, Nobel Prize, etc.). Please, remove the names of authors/co-authors from the body of manuscript - paragraphs 3.1, 3.2, 3.3, 3.4
Round 2
Reviewer 2 Report
Comments and Suggestions for Authors
I considered this journal to be at the level of a systematic review. But as I later saw, Biomolecules publishes Articles, Reviews, Communications and Editorials.
For Figure 1 it is good to mention that you used Adobe Illustrator software and because the drawing is edited with information from other articles it is good to mention these references.
Reviewer 3 Report
Comments and Suggestions for Authors
The author already addressed my concerns.
Reviewer 4 Report
Comments and Suggestions for Authors
The manuscript requires minor language corrections:
Please, check the language of manuscript for errors such as e.g.
lacking dash in combination of "drug, substance, action, etc." with "induced"
i.e. instead of "x induced" please write "x-induced" - obesity-induced, glutamate-induced...
lines 288-292 correct the sentences starting with "utilized data" - meaning is not clear
Comments on the Quality of English Language
Please, check the language of manuscript for errors such as e.g.
lacking dash in combination of "drug, substance, action, etc." with "induced"
i.e. instead of "x induced" please write "x-induced" - obesity-induced, glutamate-induced...
lines 288-292 correct the sentences starting with "utilized data" - meaning is not clear
